# Multiphase superconductivity in PdBi$_2$

Lewis Powell [1,6] ✉, Wenjun Kuang[1,5,6], Gabriel Hawkins-Pottier[1,6], Rashid Jalil[1], John Birkbeck [1], Ziyi Jiang[1], Minsoo Kim[1], Yichao Zou [2], Sofiia Komrakova [1], Sarah Haigh [2], Ivan Timokhin[1], Geetha Balakrishnan [3], Andre K. Geim[1,4], Niels Walet [1], Alessandro Principi [1] ✉ & Irina V. Grigorieva [1,4] ✉

Unconventional superconductivity, where electron pairing does not involve electron-phonon interactions, is often attributed to magnetic correlations in a material. Well known examples include high-$T_c$ cuprates and uranium-based heavy fermion superconductors. Less explored are unconventional superconductors with strong spin-orbit coupling, where interactions between spin-polarised electrons and external magnetic field can result in multiple superconducting phases and field-induced transitions between them, a rare phenomenon in the superconducting state. Here we report a magnetic-field driven phase transition in β-PdBi$_2$, a layered non-magnetic superconductor. Our tunnelling spectroscopy on thin PdBi$_2$ monocrystals incorporated in planar superconductor-insulator-normal metal junctions reveals a marked discontinuity in the superconducting properties with increasing in-plane field, which is consistent with a transition from conventional (s-wave) to nodal pairing. Our theoretical analysis suggests that this phase transition may arise from spin polarisation and spin-momentum locking caused by locally broken inversion symmetry, with p-wave pairing becoming energetically favourable in high fields. Our findings also reconcile earlier predictions of unconventional multigap superconductivity in β-PdBi$_2$ with previous experiments where only a single s-wave gap could be detected.

Superconductivity in materials with spin-dependent correlations, either ferro- or antiferromagnetic, is often found to be unconventional[1–7]. Here, Cooper pairs are bound together not by the conventional electron-phonon interaction (s-wave pairing) but by other mechanisms, typically related to these materials' intrinsic magnetism. Furthermore, coupling between an external magnetic field and spin-polarised electrons can lead to a multiplicity of superconducting phases that exist in different regions of the temperature-magnetic field phase diagram[1–4]. Such multiphase superconductivity is found in some heavy-fermion superconductors[3,5–8], as well as in liquid He-3[1,9]. Complex phase diagrams and unconventional superconductivity are also predicted in materials with strong spin-orbit coupling (SOC) due to spin-momentum locking, but the nature of superconductivity in this case is much less explored, especially experimentally. Recent findings include topologically protected surface superconductivity[10], superconducting diode effect[11] and critical fields greatly exceeding the paramagnetic Pauli limit[12–16]. An emergence of phase transitions is another enticing possibility, e.g., an external magnetic field can distort the spin-locked Fermi surfaces, changing the energy balance between superconducting phases with different pairing symmetries[17]. However, the even-parity (spin-singlet) and odd-parity (spin-triplet) superconducting order parameters in such materials are typically mixed[18,19] and phase transitions in the superconducting state are rare, with recently reported heavy fermion CeRh$_2$As$_2$[20–22] and Li-intercalated bilayer MoS$_2$[23] the only known examples. In addition, the complexity of competing interactions allows alternative interpretations of the

[1]Department of Physics and Astronomy, University of Manchester, Manchester, UK. [2]Department of Materials, University of Manchester, Manchester, UK. [3]Department of Physics, University of Warwick, Coventry, UK. [4]National Graphene Institute, University of Manchester, Manchester, UK. [5]Present address: National Innovation Institute of Defense Technology, AMS, Beijing, China. [6]These authors contributed equally: Lewis Powell, Wenjun Kuang, Gabriel Hawkins-Pottier. ✉e-mail: lewis.powell@manchester.ac.uk; alessandro.principi@manchester.ac.uk; Irina.V.Grigorieva@manchester.ac.uk

                                                                    

experimentally observed phase diagrams[21] and further studies on different experimental systems are needed to unravel the mechanisms underpinning the effect of the magnetic field.

Here, we report a magnetic-field-driven phase transition within the superconducting state of the layered tetragonal superconductor β-PdBi$_2$. Its basic superconducting properties were described in the literature already in the 1950s[24,25]. More recently, theory identified it as a candidate topological superconductor, where multiple superconducting gaps with different symmetries are expected to open on different Fermi surfaces[26–29]. Band structure calculations[26], angle-resolved photoemission spectroscopy (ARPES)[27,30] and quasiparticle interference imaging (QPI)[28] found spin textures both in the surface and bulk electronic bands. Yet, only single-gap $s$-wave superconductivity could be detected so far in a range of different experiments[31–33], with only one recent neutron scattering study indicating a possibility of two gaps of different magnitude and/or momentum-dependent gap anisotropy[34]. In our work, we used tunnelling spectroscopy on PdBi$_2$-hBN-few-layer-graphene heterostructures and resistance measurements on exfoliated crystals, and found evidence of a magnetic-field driven phase transition: While at low fields $B$, both in-plane and out-of-plane, β-PdBi$_2$ behaves as a conventional $s$-wave superconductor, at in-plane $B$~0.1-0.2 T we observed a transition to a new superconducting state with characteristics of unconventional pairing and a nodal gap. This is seen as a sharp change in the characteristics of the tunnelling spectra above and below a transition field $B^*$ (zero-bias conductance, extracted gap value, pair-breaking strength), as well as a kink in the phase diagram $B_{c2}(T)$ for the in-plane field.

## Results

Importantly for the present study, we were able to grow high-quality single crystals of the tetragonal β-PdBi$_2$, see X-ray diffraction in Supplementary Fig. 1b. The high crystal quality is further confirmed by the exceptionally low hysteresis in the DC magnetisation, $M(B)$, and the sharpness of the superconducting transition, Fig. 1e and Supplementary Fig. 1a. The superconducting coherence length $\xi$ and magnetic field penetration depth $\lambda$ for our PdBi$_2$ were determined from the critical fields $B_{c1}$ and $B_{c2}$ obtained from magnetisation measurements (Methods), yielding $\xi_{ab}(0) \approx 22$ nm (in-plane coherence length), $\xi_c(0) \approx 17$ nm (out-of-plane) and $\lambda(0) \approx 240$ nm. The in-plane coherence length is only slightly shorter than the low-temperature mean free path $l \approx 25$ nm determined from magnetoresistance measurements on relatively thick (~1 μm thick) crystals from the same batch.

To measure the superconducting gap $\Delta$, we have fabricated superconductor-insulator-normal metal (SIN) tunnel junctions, where thin slabs of PdBi$_2$ mechanically exfoliated from a bulk crystal (such as shown in Supplementary Fig. 1b) served as the superconducting electrode. Few-layer graphene (FLG) and 2-3 layer thick hBN were used as the normal electrode (N) and the insulating barrier (I), respectively. Figure 1b,c shows an optical image and a schematic of a typical device (see "Methods" for fabrication details). Cross-sectional transmission electron microscopy on one of the used devices ("Methods" and Fig. 1d) verified that the fabrication process did not induce any phase transformations, nor introduced defects. The design allowed us to measure both the tunnelling conductance and the resistance in Hall bar geometry on the same device. Five devices were studied, containing PdBi$_2$ crystals with thicknesses between 50 and 140 nm. They all showed the same qualitative behaviour but quantitative characteristics (critical temperature $T_c$, upper critical field $B_{c2}$) were found to depend on the thickness $d$ and were different from those for bulk crystals (10–100 μm thick). While it would be interesting to study even thinner crystals, unfortunately, we found it impossible to exfoliate crystals with $d$<50 nm and lateral dimensions sufficient for device fabrication (see "Methods" for details). Typical examples described in detail below are for devices with $d$ = 80 nm (device A) and $d$ = 50 nm (device B).

The differential tunnelling conductance $G(V_b) = dI(V_b)/dV$ was measured by applying a small AC excitation $dV \sim 50$ μV superimposed on a DC bias voltage $V_b$ and detecting the AC current $dI$ between the top (FLG) and bottom (PdBi$_2$) electrodes. At low $T$ and zero $B$, we observed spectra typical for conventional SIN tunnel junctions[35], with a full gap seen as zero conductance for $V_b$ < 0.2 mV, and sharp conductance peaks just above the gap (Fig. 1f). To quantify $\Delta$ as a function of $T$, we fitted the measured tunnelling conductance $G_{NS}(V_b)$ using the standard expression[35]

$$G_{NS} = \frac{dI}{dV} = \frac{G_{NN}}{N_N(0)} \int_{-\infty}^{+\infty} N_S(E, \Gamma, \Delta) \frac{\partial f(E + eV_b, T)}{\partial(eV_b)} dE, \quad (1)$$

where $G_{NN}$ corresponds to both electrodes being in the normal state, $N_N(0)$ and $N_S(E, \Gamma, \Delta)$ are the density of states (DoS) at the Fermi level for the superconducting electrode in the normal and superconducting state, respectively; $f(E + eV_b, T)$ the Fermi-Dirac distribution, $E$ the quasiparticle energy and $\Gamma$ the quasiparticle lifetime broadening parameter. The superconducting DoS is given by the Dynes formula[36]

$$\frac{N_S(E, \Gamma, \Delta)}{N_N(0)} = \text{Re}\left[\frac{E - i}{\sqrt{(E - i)^2 - \Delta^2}}\right] \quad (2)$$

Figure 1e shows the conductance map and the order parameter $\Delta(T)$ extracted from individual spectra, such as shown in Fig. 1f. Here $\Delta(T)$ is well described by the universal formula for the BCS gap, $\Delta(T) = 1.76 k_B T_{c0} \tanh(1.74\sqrt{T_{c0}/T - 1})$, indicating standard $s$-wave superconductivity at $B = 0$.

In contrast, the evolution of the tunnelling spectra with a magnetic field is highly unusual. Firstly, there is a large anisotropy between the in-plane and out-of-plane $B$. This is seen qualitatively in the conductance maps of Fig. 2a, b: In the out-of-plane field, $B^\perp$, the gap – indicated approximately by the width of the low $G/G_0$ region (brown area of the maps) – decreases smoothly, and the evolution of the individual spectra is qualitatively similar to their evolution as a function of temperature (c.f. Fig. 1f and Supplementary Fig. 2a). However, for in-plane field, $B^\parallel$ (Fig. 2a) there are two distinct regions in the $B$ dependence: below ~0.2T the gap is rapidly suppressed until a pronounced 'kink' appears at $B^* \approx 0.2$ T, after which it decreases slowly up to $B_{c2} \approx 1.6$T. In addition, distinct responses above and below $B^*$ are seen in individual spectra, not only in the spectral shape and zero-bias conductance (ZBC), where qualitative changes are clear in Fig. 2c, but also in the evolution of the parameters describing the spectra, the order parameter $\Delta(B)$ and the pair-breaking strength $\zeta$ (see below for a detailed discussion). In terms of the field sweep direction, with our experimental accuracy, there was no discernible difference in the tunnelling spectra measured in an increasing/decreasing field.

The anisotropy is further evidenced in the phase diagram, Fig. 2d, which compares the $T$ dependence of the in-plane and out-of-plane upper critical field, $B_{c2}$. In contrast to a smooth linear increase of $B_{c2}^\perp(T)$, as is typical for thin films[35], $B_{c2}^\parallel(T)$ shows a clear kink and two distinct regions on the phase diagram. Here we used measurements of the resistance $R(T, B)$ on the same exfoliated PdBi$_2$ crystals in the Hall bar configuration (see schematic in Fig. 1c), taking the values of $B$ at which the resistance is 90% of the normal-state value (just below the transition to the superconducting state) as $B_{c2}$. As the $R(B)$ curves for the in-plane field were always sharp, varying this criterion had little effect on the extracted $B_{c2}$ and did not change the two trends. We note that the sharpness of the resistance curves at all $T$ (Fig. 2e) is consistent with the absence of vortices, as both $B$ and $\Delta$ are essentially uniform over the crystal thickness $d \sim (2-3)\xi \ll \lambda$. For comparison, Supplementary Fig. 2b shows $R(T, B^\perp)$ the out-of-plane field, where the presence of vortices broadens the resistance curves, particularly at low $T$.

We first analyse the observed tunnelling conductance for in-plane $B$ in light of the known evolution of the DoS for an $s$-wave

                                                                    

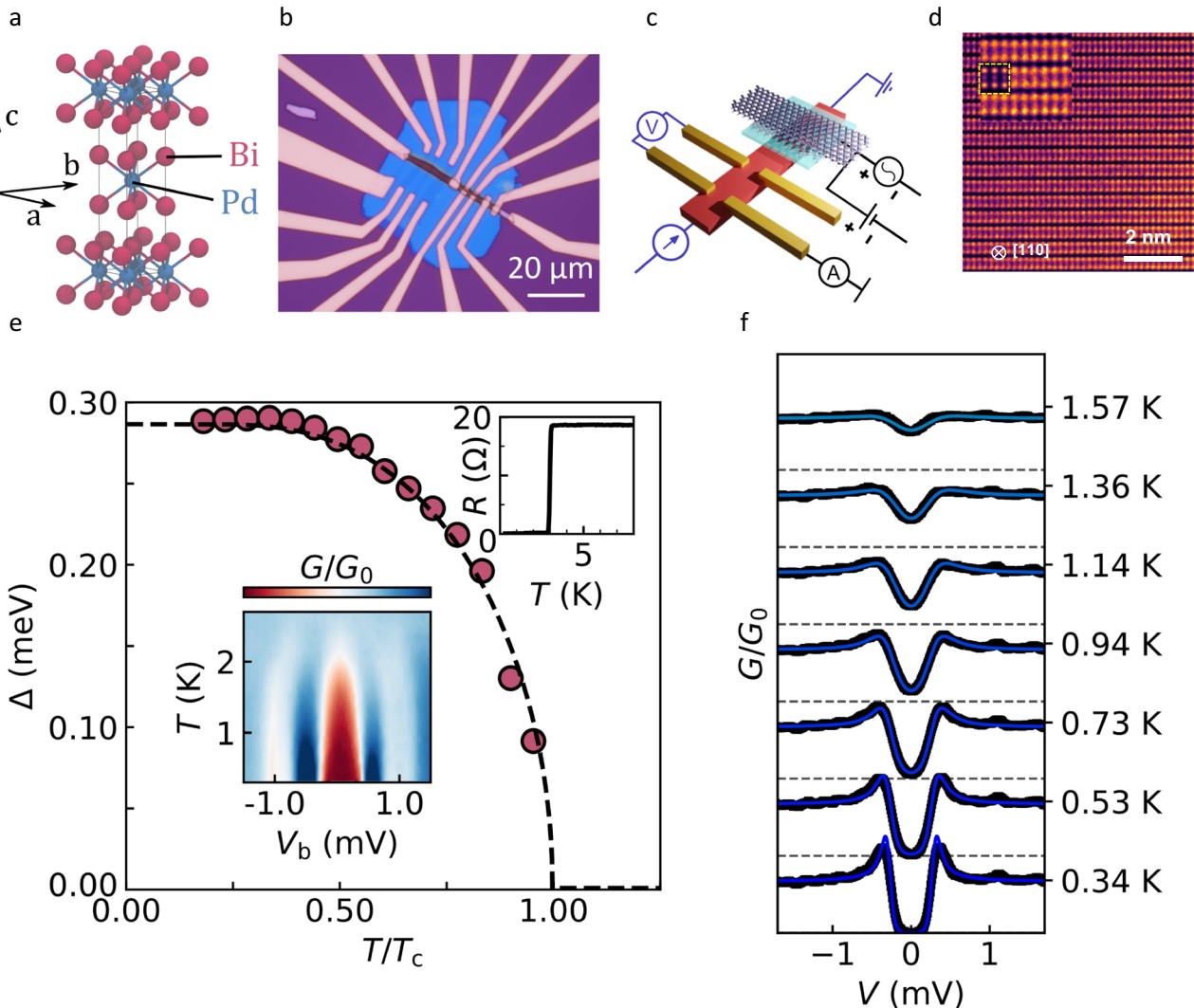

**Fig. 1 | Device design and tunnelling characteristics of β-PdBi₂. a** Crystal structure of β-PdBi₂. Bismuth atoms within each Bi-Pd-Bi layer are arranged tetragonally around Pd to form a square Bi bilayer. Neighbouring layers are staggered in an AB configuration. **b** Optical image of one of our devices. **c** Schematic of a device combining a tunnelling conductance measurement scheme (black) and four-probe contact configuration for resistance measurements (blue). For tunnelling, few-layer graphene (ball and stick model) acts as the normal metal, PdBi₂ is the superconducting electrode (red), and 2-3 layer insulating hBN is used as a tunnel barrier (cyan). The entire structure is encapsulated in 100 nm hBN, not shown here. **d** Atomic-resolution cross-sectional STEM image of a thin slice lifted from a device

after completing the tunnelling measurements ("Methods"). The inset shows a zoomed-in section of the main image overlapped with the simulated HAADF image (the latter outlined by the yellow dashed line). **e** *Main panel*: Temperature-dependent superconducting gap, $\Delta(T)$, extracted from fitting individual tunnelling spectra for device B (symbols). The dashed line is fit for weak-coupling BCS theory. *Insets*: Superconducting transition in $R(T)$ (top) and a zero-field tunnelling conductance map (bottom). **f** Zero-field tunnelling spectra at different $T$, see labels. The spectra (black symbols) are accurately described by the standard Dynes model (solid blue lines). Data for device B. Source data are provided as a Source Data file.

superconductor. For a qualitative comparison, we modelled the tunnelling spectra using Maki's solutions of the generalised Gorkov equation for a thin film in parallel field[37–40], i.e., such that both $B$ and $\Delta$ are uniform across the film's thickness. (Assuming that the coherence length $\xi$ and penetration depth $\lambda$ in exfoliated PdBi₂ crystals are approximately the same as in our bulk samples, this condition is satisfied for all our devices having $d \lesssim 100$ nm.) As illustrated in Supplementary Fig. 3a, the modelled spectra have the following qualitative features: (i) a full gap of decreasing size persists up to $B$ close to $\sim 0.7B_{c2}$, with ZBC remaining zero; (ii) only at $B > 0.7B_{c2}$ does superconductivity become gapless, with a rapid increase in ZBC; (iii) quasiparticle peaks are almost fully suppressed by a relatively low $B \sim 0.5B_{c2}$, while the spectra remain fully gapped. All these features have been reported in the literature for the tunnelling spectra of conventional superconductors (Sn, Sn-In[38,39]), and they are also seen in our experiment in the low-field region, $B^{\parallel} \lesssim B^{*}$, see Fig. 2c and

Supplementary Figs. 4 and 5a. In contrast, above $B^{*}$, the spectra become "V"-shaped, and ZBC increases rapidly, indicating the presence of low-energy quasiparticle excitations inside the gap[41,42]. The latter observation is particularly unusual as it indicates *gapless* superconductivity over a wide range of $B \ll B_{c2}$, in stark contrast to the conventional behaviour but consistent with nodal superconductivity[41,42]. More subtle differences between the two regions of the magnetic field are seen in the spectral peaks corresponding to quasiparticle excitations just above the gap: below $B^{*}$, their height is rapidly suppressed by increasing $B^{\parallel}$, as expected for a conventional superconductor, but no further suppression is seen above $B^{*}$ and they remain prominent up to $B \sim 0.5B_{c2}$ (Fig. 2c).

More quantitatively, Maki theory[37,43] allows using tunnelling conductance to evaluate the two parameters that describe the effect of in-plane magnetic field on superconductivity: the order parameter $\Delta(B^{\parallel})$ and the depairing strength $\zeta(B^{\parallel})$ due to time-reversal-breaking

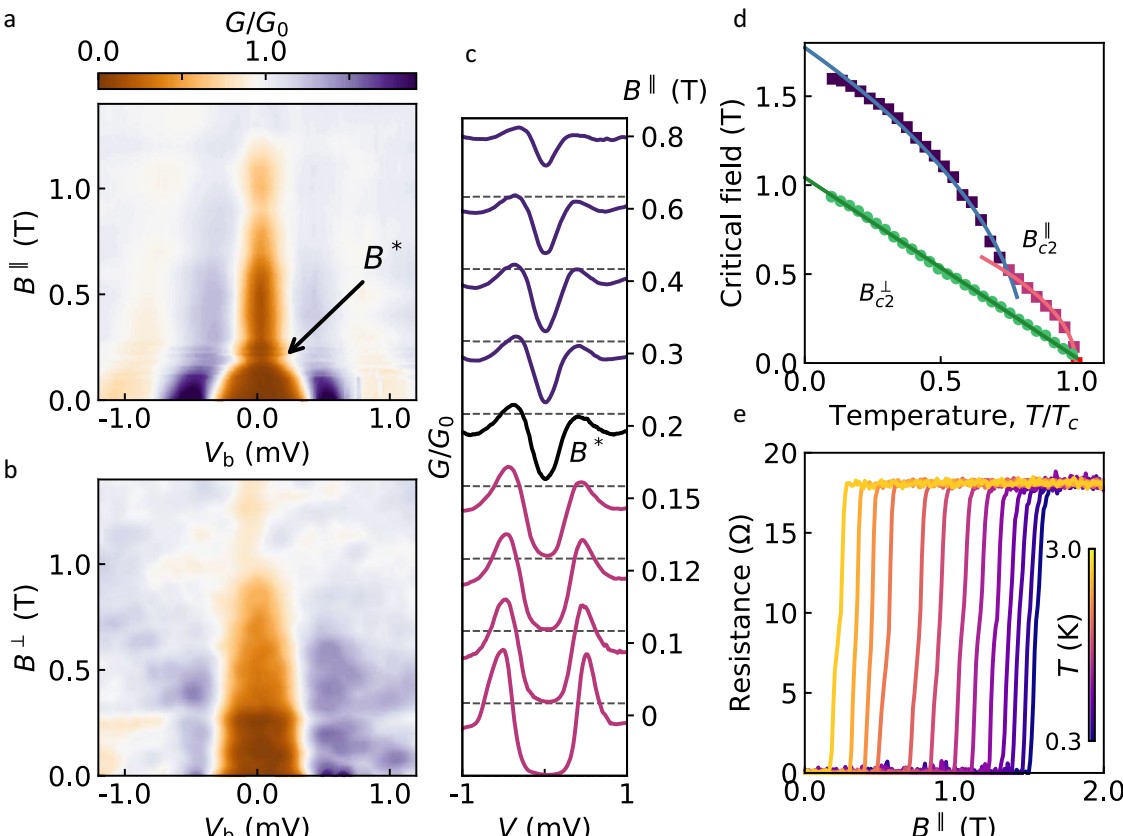

**Fig. 2 | Phase transition under in-plane magnetic field. a, b** Maps of the normalised tunnelling conductance for in-plane (**a**) and out-of-plane (**b**) magnetic fields. $T = 0.3$ K. $G_0$ corresponds to both electrodes being in the normal state. Brown areas correspond approximately to the spectral gap. The arrow in (**a**) indicates the transition field $B^*$ (see text). **c** Selected spectra from (**a**) emphasising the change in spectral shape at $B^*$. Values of $B$ are shown as labels. Dashed horizontal lines correspond to $G/G_0 = 0$. **d** Temperature-dependent upper critical fields for in-plane and out-of-plane $B$ extracted from $R(B)$ measurements, such as shown in (**e**). Shown values of $B_{c2}$ correspond to $R = 0.9R_N$, where $R_N$ is the normal state resistance. Green circles show $B_{c2}^\perp$, red squares $B_{c2}^\parallel$ below the kink at 0.5 T, and blue squares $B_{c2}^\parallel$ above 0.5 T. Solid lines are fits to Eqs. (4) and (5), see text. **e** Resistance vs in-plane magnetic field at different $T$. Data for device A. Source data are provided as a Source Data file.

perturbations that split Cooper pairs, see Eqs. (9)–(12) in Methods. The theory is known to accurately describe $\Delta(B^\parallel)$ and $\zeta(B^\parallel)$ in conventional superconductors such as Sn[38,39]. We emphasise that no further fitting parameters – beyond $\Delta(B^\parallel)$ and $\zeta(B^\parallel)$ – are needed to describe the tunnelling spectra for a material with a relatively low $B_{c2}$, such as PdBi$_2$ ("Methods"). In fact, the majority of the changes are due to the pair-breaking effect of the magnetic field (compare Supplementary Fig. 3b and 3c). Other quantities appearing in Eqs. (9)–(12) are fixed: $T$ is the experimental temperature (0.3 K or $0.1T_c$), and the g-factor, known to be $g \approx 2$ for β-PdBi$_2$, fixes the Zeeman energy $\mu_B B$ (here $\mu_B$ is the Bohr magneton).

Figure 3a–c and Supplementary Fig. 5a demonstrate that the theory provides a good fit to the low-$B$ spectra for our PdBi$_2$, $\Delta(B^\parallel < B^*)$ and $\zeta(B^\parallel < B^*)$, which is further evidence that below $B^*$ its superconductivity is underpinned by conventional s-wave pairing. In contrast, at $B^\parallel > B^*$ the fits to the Maki theory become poor, and the fast-increasing ZBC cannot be described by any realistic value of the depairing strength $\zeta$, see Fig. 3c and Supplementary Fig. 5b. Persistence of the quasiparticle peaks well beyond $B^*$ is also contrary to the theory expectations (compare modelling in Supplementary Fig. 3a with experimental spectra in Figs. 2c and 3b, and Supplementary Figs. 4 & 5b).

As for the effect of the out-of-plane field, no kink in $G(V_b, B_\perp)$ or other evidence of a field-induced phase transition could be seen in any of our devices (see Supplementary Fig. 2a for an example), and we, therefore, conclude that superconductivity, in this case, remains s-

wave. We note, however, that the spectra in an out-of-plane field are not directly related to DoS and cannot be fitted by any simple model, as vortices can contribute to the mid-gap conductance.

We now show that all the above field-induced changes in the superconducting characteristics of our PdBi$_2$ can be explained by a transition from conventional s-wave pairing to nodal superconductivity, with anisotropic p-wave pairing being the most likely candidate in the latter case. An excess ZBC and a Dirac-like sub-gap DoS are well-known characteristics of a superconducting gap vanishing at lines on the Fermi surface[41,42]. In our devices, ZBC increases sharply (approximately linearly) as soon as $B$ exceeds $B^*$, Fig. 3d. For a more quantitative relationship, we fitted the high-field spectra using DoS for a nodal p-wave order parameter $\Delta_{\hat{k}} = \Delta \cos \theta_k$, and including a 'broadening' parameter $\Gamma$ due to pair-breaking effects (see Supplementary Note 2.1 for a derivation):

$$\frac{N_S(E, \Gamma, \Delta)}{N_N(0)} = \text{Re}\left[\frac{E + i\Gamma}{\Delta} \arcsin\left(\frac{\Delta}{E + i\Gamma}\right)\right] \quad (3)$$

As shown in Fig. 3b and Supplementary Fig. 5b, the nodal DoS fits the spectra accurately at all $B > B^*$, whilst the fits with the Maki model are poor even if we allow the pair-breaking strength $\zeta$ to take on unrealistically high values (see Supplementary Fig. 5b for a detailed explanation). Given that below $B^*$ the tunnelling spectra are accurately described by the Maki theory, we use Eqs. (9)–(12) ("Methods") to extract the order parameter $\Delta(B)$ at $B \leq B^*$ and for $B > B^*$ use Eq. (3). The

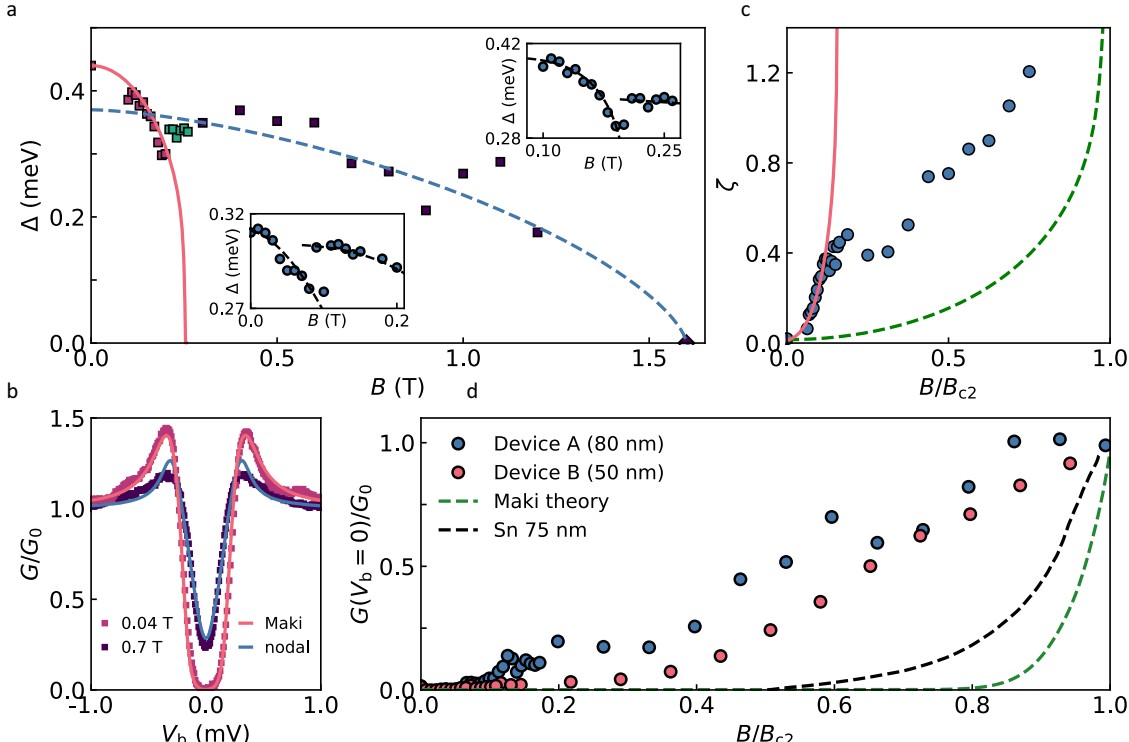

**Fig. 3 | Superconducting gap of PdBi$_2$ for in-plane magnetic field. a** *Main panel*: Superconducting order parameter Δ extracted from fits to individual spectra vs in-plane magnetic field $B^{\parallel}$. Data for device A ($d \approx 80$ nm), $T = 0.3$ K. Red squares correspond to $\Delta(B^{\parallel})$ extracted using the Maki theory and BCS DoS, dark-blue squares to nodal DoS (see text), and green squares are fit to the Maki theory in the intermediate region. The red solid line is fit for Eqs. (9, 10) yielding an apparent critical field for low-$B$ s-wave superconductivity $B_c^{s-wave} \approx 0.25$T. The Dashed blue line is a guide to the eye. *Insets*: Detailed view of $\Delta(B^{\parallel})$ in the low-$B$ region for device A (top inset) and device B (bottom inset). A pronounced kink in $\Delta(B^{\parallel})$ seen for both devices, corresponds to suppression of the s-wave gap followed by the appearance of a new order parameter, as emphasised by the dashed lines (guides to the eye). **b** Representative spectra at $B<B^*$ (red squares) and $B>B^*$ (dark-blue squares) revealing the change from conventional (s-wave) to nodal gap. Solid lines are fits to the two models, see legend. Details of fitting are explained in Methods. **c** Evolution

of the depairing strength parameter $\zeta(B^{\parallel})$ extracted from fitting of the experimental spectra to the Maki theory (blue symbols). Below $B^*$, $\zeta$ follows the expected behaviour for an s-wave superconductor with a critical field $B_c^{s-wave} = 0.25$ T; this is shown by the red solid line calculated using Eqs. (11), (12) ("Methods"). The dashed green line shows $\zeta(B^{\parallel})$ calculated using the same equations for an s-wave superconductor with $B_{c2} = 1.6$ T (actual upper critical field for our PdBi$_2$). Attempting to apply the Maki theory above $B^*$ (as detailed in Supplementary Fig. 5b) results in unphysically large values of $\zeta$; this is clear from a comparison between the extracted $\zeta$ (blue symbols) and the theory prediction (dashed green line). **d** Evolution of zero-bias conductance with $B^{\parallel}$ for device A (blue) and device B (red). For comparison, dashed lines show corresponding results predicted by theory[37] (green) and experimental data for a conventional BCS superconductor (75 nm Sn film) taken from ref. [39] (black). Source data are provided as a Source Data file.

results are shown in Fig. 3a. Similar to the suppression of the superconducting gap seen qualitatively in the conductance map in Fig. 2a, at low $B$ the order parameter is rapidly suppressed, tending towards an extrapolated critical field ≈ 0.25 T (red curve in Fig. 3a). At $B^* \approx 0.2$T, Δ is seen to increase again, in agreement with the 'kink' seen in raw $G(V_b, B^{\parallel})$ in Fig. 2a, and at $B > 0.5$T it starts to decrease slowly towards $B_{c2} \approx 1.6$T. The spectra in the transitional region between $B \sim 0.2$ and 0.4 T can be fitted equally well by Eqs. (9)–(12) and eq. (3) (i.e., no preference for either the Maki or nodal model).

The sharp suppression of the conventional s-wave order parameter followed by a transition towards a larger Δ and a slow approach towards Δ = 0 at $B_{c2}$ is seen in all our devices, the only difference being the exact value of $B^*$ which generally decreases for thinner crystals (cf. $\Delta(B)$ for $d = 50$ and 80 nm in Fig. 3a). Together with the increase in ZBC, the changes in the spectral shape and the different evolution of the pair-breaking strength $\zeta(B)$ above and below $B^*$(Fig. 3c), this implies a transition from conventional s-wave pairing to a new, field-induced, phase characterised by unconventional nodal pairing.

The phase diagram in Fig. 2d - $B_{c2}^{\parallel}(T)$ and $B_{c2}^{\perp}(T)$ - provides further insight into the effect of magnetic field on the superconductivity of β-PdBi$_2$. As shown in Fig. 2d, out-of-plane $B_{c2}^{\perp}(T)$ is accurately described by the 2D Ginzburg-Landau (GL) theory[35] at all $B$ and $T$, indicating

conventional behaviour:

$$B_{c2}^{\perp}(T) = \frac{\Phi_0}{2\pi\xi_{ab}(0)^2}\left(1 - \frac{T}{T_c}\right) \qquad (4)$$

Here $\Phi_0$ is the magnetic flux quantum and $\xi_{ab}$ the in-plane coherence length. The fit yields $\xi_{ab}(0) = 18$ nm and $B_{c2}^{\perp}(0) \approx 1$T, close to the extrapolated bulk values of about 20 nm and 0.8 T ("Methods"). In contrast, and in agreement with our findings from tunnelling spectroscopy, $B_{c2}^{\parallel}(T)$ cannot be described by a single GL fit due to a kink at $B \sim 0.5$ T, which implies that the low-$B$ superconducting phase is eclipsed by a different phase at higher $B$. Individually, both parts of the $B_{c2}^{\parallel}(T)$ curve can be described by the GL expression[35]

$$B_{c2}^{\parallel}(T) = \frac{\sqrt{12}\Phi_0}{2\pi\xi_{ab}d}\sqrt{1 - \frac{T}{T_c}}, \qquad (5)$$

with the low-$B$ fit yielding $d = 63$ nm, close to the actual thickness of the PdBi$_2$ crystal, 80 nm. While 0.5 T is notably above $B^*$ for this device, this is likely because the kink in $B_{c2}(T)$ is where the high-field superconductivity completely overtakes the s-wave phase, whereas the kink in $G(V_b)$ corresponds to its onset. Indeed the field corresponding

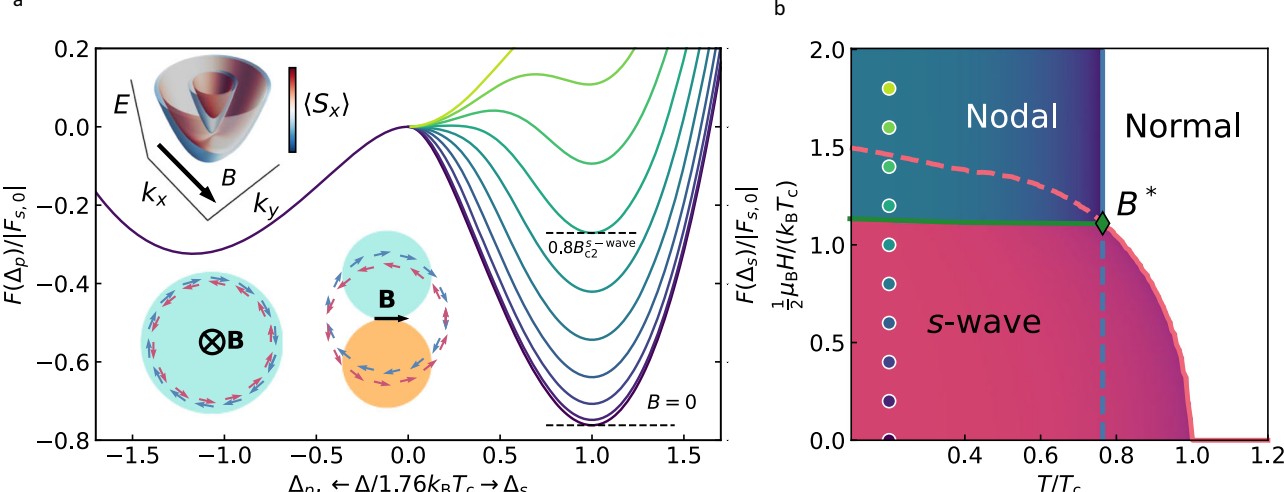

**Fig. 4 | Effect of the magnetic field on the free energy of s-wave and p-wave superconducting states of β-PdBi₂ and the corresponding phase diagram.** **a** Normalised free energy at $T = 0.1T_c$ for several values of the magnetic field (colour-coded as indicated by dots in (**b**)) as a function of two possible order parameters: p-wave $\Delta_p$ in BCS units of $\Delta/1.76k_BT_c$ to the left of 0 (negative values) and s-wave $\Delta_s$ to the right. Within our model, p-wave pairing is insensitive to the magnetic field. *Bottom-left inset:* Isotropic lift of band degeneracy due to out-of-plane magnetic field. States of opposite momenta have opposite spin polarisations, favouring s-wave coupling. *Top and bottom-right insets:* Anisotropic lift of band degeneracy due to the in-plane magnetic field. States of opposite momenta, perpendicular to the direction of **B**, have parallel spin polarisation, favoring p-wave coupling. **b** Phase diagram constructed by minimising the free energy $F(\psi, \eta, B, T)$ for the s-wave order parameter $\psi$ and a spin-polarised triplet order parameter $\eta$. Transition to the nodal p-wave state occurs at a temperature-independent field $B^*$. Coloured dots indicate the values of $B$ corresponding to the free energy curves shown in (**a**). Source data are provided as a Source Data file.

to the kink in $B_{c2}(T)$ in Fig. 2d is close to $B$ corresponding to the maximum for the nodal order parameter in Fig. 3a.

The above results, using different experimental probes, show that only a single s-wave gap is present in zero- and out-of-plane $B$, in agreement with previous studies[28,31]. This suggests that any unconventional pairing, in this case, is either energetically unfavourable or obscured by a larger BCS gap, whereas a sufficiently strong in-plane $B$ favour pairing in the unconventional channel. This was hinted at in a recent neutron scattering experiment[34] where an unusual $T$ dependence of the superfluid density indicated unconventional pairing in $B^{\parallel}$ but not in $B^{\perp}$.

## Discussion

To explain the observed field-induced transition in the superconducting state, we have constructed a minimal tight-binding model ("Methods" and Supplementary Note 2.2) taking into account the 'hidden' symmetry breaking[17,28,29] in β-PdBi₂: Even though the atomic arrangement in these crystals is globally centrosymmetric, electrons at Bi sites in neighbouring layers experience a locally non-centrosymmetric environment, which gives rise to a Rashba-like SOC[44,45] and in-plane spin-momentum locking with opposite spin polarisations, see the sketch in Fig. 4a. Because of the spin polarisation, an in-plane magnetic field produces anisotropic splitting of the energy bands, which is large where **B** is aligned with the spin-orbit field (for momenta orthogonal to **B**), Fig. 4a, and very small where the two are perpendicular. As a result, opposite-**k** states have parallel spin polarisations and can be expected to favour equal-spin (triplet) pairing with nodes along the direction of **B**.

Figure 4a shows the free energy $F(\psi, \eta, B, T)$ calculated using separate gap equations for each Zeeman-split band for two pairing channels (with the s-wave order parameter $\psi$ and a spin-polarised triplet order parameter $\eta$), see Methods and Supplementary Notes 2.3–2.5 for details. At low $B$, the s-wave state has lower energy and superconductivity is gradually suppressed by the magnetic field. However, as $B$ increases above a critical value (~ $0.7\,B_{c2}^{s-wave}$ for the interaction parameters in Fig. 4a), the free energy of the s-wave state becomes larger than for the triplet state, resulting in a first-order phase

transition to the triplet state. By minimising the free energy $F(\psi, \eta, B, T)$ for $\psi$ and $\eta$, we have obtained the ground state mean-field configuration at each $(B, T)$ and constructed a phase diagram (Fig. 4b) which shows that a transition from s-wave to triplet state occurs at the $T$-independent transition field $B^*$. The value of $B^*$ depends on the parameters of the model, in particular, on the interaction parameters $U$ and $V$ for s-wave and p-wave pairing, respectively. In general, as $V$ increases at fixed $U$, the transition to nodal superconductivity occurs at a lower $B$. In the limit $V \gg U$, $B^*$ vanishes and nodal superconductivity is favourable at all magnetic fields. The values of $U$ and $V$ for β-PdBi₂ can be estimated from the experimental critical temperatures $T_c^{s-wave}$ and $T_c^{p-wave}$. For the ~ 80 nm thick PdBi₂ crystal, we found $T_c^{s-wave} \approx 3K$ (Fig. 1e) and $T_c^{p-wave} \approx 2.4K$ (here $T_c^{p-wave}$ is taken as an extrapolation to $B = 0$ in Fig. 2d). Using this result and the fact that $T_c$ of a singlet superconductor is given by

$$k_B T_c = 1.13\hbar\omega_D e^{-\frac{1}{N_0 U}} \tag{6}$$

we estimate $UN_0 = 0.26$. Here $N_0$ is the DoS per unit cell volume, and the Debye frequency $\hbar\omega_D = 0.01$ eV is taken to be of the order of the largest phonon frequency for β-PdBi₂[46]. The value of $V$ can be estimated in a similar way, by replacing $U$ in Eq. (6) with $2^{-1}V\rho^2/(1+\rho^2)$, where $\rho = \alpha k_F/\epsilon$ and $k_F = \sqrt{2}(\mu m + \alpha^2 m^2 - (\alpha^4 m^4 + 2\alpha^2 m^3 \mu + \epsilon^2 m^2)^{1/2})^{1/2}$ is the Fermi wavevector of the 2D-like band for effective mass $m$, chemical potential $\mu$, Fermi energy $\epsilon$ and Rashba SOC strength $\alpha$. Using the same $\hbar\omega_D$ as above, we estimate $VN_0 = 0.78$, i.e., a similar order of magnitude as $U$, as can be expected for a realistic superconductor.

Our experimental observations are in agreement with the predicted first-order phase transition from s-wave to triplet nodal pairing. A suppression (but not closing) of the s-wave gap in Fig. 3a is consistent with the appearance of nuclei of a new phase (nodal pairing) at $B^*$, as expected for a first-order phase transition. Furthermore, the apparent critical field for the s-wave phase (referred to as $B_c^{s-wave} \approx 0.25$T in Fig. 3) is larger than $B^*$ because the latter corresponds to the appearance of the first nuclei of the new phase, while the superconductor becomes 'fully p-wave' only at fields well above $B^*$. In the example of

Fig. 3a, the new phase takes over at $B \sim 0.4$ T or $\sim 2B^*$. In the intermediate field range, the two phases coexist, which can also explain why the tunnelling spectra here are described equally well by both the $s$-wave and the nodal order parameters. Let us emphasise that it would be incorrect to treat $B_c^{s-\text{wave}} \approx 0.25$ T in Fig. 3 as a true critical field for $s$-wave superconductivity, because of the existence of the transition to $p$-wave pairing. Rather, the interaction between the two pairings for the in-plane field suppresses $s$-wave superconductivity at a much lower $B$ that what would be the case without it (or what is observed in the out-of-plane $B$, where the material remains superconducting up to $\sim 1$ T, Fig. 2d).

We note that the free energy in Fig. 4 does not include orbital depairing, leading to an unphysical result that $T_c^{p-\text{wave}}$ above $B^*$ is independent of the magnetic field and $B_{c2}^{p-\text{wave}}$ diverges. In addition, $B_{c2}^{s-\text{wave}}$ is equal to the Pauli paramagnetic limit (Fig. 4b), much higher than the experimental values, where orbital depairing dominates. Nevertheless, assuming that both $s$-wave and $p$-wave superconductivity are suppressed by orbital depairing in a similar way, our model correctly captures the fact that the transition to nodal pairing (for realistic parameters $U$ and $V$) occurs well below $B_{c2}^{s-\text{wave}}$, as observed experimentally. Finally, we note that triplet pairing is unfavoured when the magnetic field is out-of-plane. This agrees with the experiment, where the pairing transition is induced by the in-plane, but not out-of-plane $B$.

The bilayer Rashba model discussed above and described in detail in Supplementary Notes 2.2–2.4 is the simplest model that reproduces all main features of the $\beta$-PdBi$_2$ band structure, including the nontrivial spin helicity of the bulk and surface states. In turn, a $p$-wave triplet phase is the simplest nodal phase that emerges in this model and triggers a phase transition that can explain the experimental observations. Electronic states of a cylindrical Fermi surface can be described by a continuum Hamiltonian. We then assume two competing interactions: a local (Hubbard) and a non-local one that couples electrons sitting on nearest-neighbouring sites that belong to different layers. The only allowed pairing channels here are nodeless $s$-wave states, an odd-parity pair density wave (PDW) state that changes sign each sublayer, and the $|m_L| = 0$ and $|m_L| = 1$ spin-triplet states[47]. While the PDW and $|m_L| = 0$ triplet phases have point nodes at $k_x = k_y = 0$ and are, therefore, not compatible with the experimental spectra[41], the only ones that host nodal lines are components of the $|m_L| = 1$ triplet pairing. This effective $p$-wave triplet phase is, therefore, the simplest nodal phase that can exist in this system. We note that higher symmetry $d$-wave states can also have nodal lines, but they are not compatible with the type of interaction assumed in our simple model, and one would have to consider more contrived non-local interactions to stabilise such phases. Let us also emphasise the role of the in-plane magnetic field: It can induce a substantial modification of the band structure in $\beta$-PdBi$_2$ because of the spin-momentum locking, which itself is due to the strong spin-orbit coupling. In turn, the changes in the band structure make the effective $p$-wave triplet phase more stable than the $s$-wave above a transition field, a result that is compatible with experimental observations. This dependence on the applied magnetic field also suggests that a singlet $d$-wave pairing may not be suitable to describe the observed transition because a Zeeman field that only affects the spin would not distinguish between two singlet states, or favour one over the other.

The above discussion did not include our puzzling observation that the superconducting parameters ($T_c$, $\Delta$, $B_{c2}^{\parallel}$) of PdBi$_2$ crystals with $d \leq 140$ nm are strongly dependent on $d$, with $T_c$ decreasing from 3.6 K to 1.8 K as $d$ is reduced from 140 to 50 nm, see Supplementary Fig. 6. In contrast, $B_{c2}^{\parallel}$ for these crystals is notably enhanced compared to the bulk ($d \sim 100$ μm) to approximately $B_{c2}^{\parallel}(0) \approx 1.6$ T (Fig. 2d) vs bulk $B_{c2}^{\parallel}(0) \approx 0.9$ T (Supplementary Fig. 1a), corresponding to a strong enhancement of $B_{c2}^{\parallel}/T_c$ ratio or superconductivity becoming more robust against $B^{\parallel}$ for thinner crystals. Also, the transition field $B^*$ appears to show a thickness dependence: it is about twice lower for

50 nm thick PdBi$_2$ compared to 80 nm, while two devices of a similar thickness showed similar $B^*$. No thickness dependence could be detected for the out-of-plane field. As $T_c(d)$ for our thin crystals accurately follow the $1/d$ dependence (Supplementary Fig. 6), this implies that the order parameter must be modified near the surface[48], with the surface contribution increasing for thinner crystals. Surprisingly, in the case of PdBi$_2$, the effect sets in at ~50 times larger thicknesses compared to thin films of conventional superconductors (~ 100 nm vs 2–5 nm)[48,49], indicating a different underlying mechanism. A possible explanation for the suppression of $T_c$, the relative enhancement of $B_{c2}^{\parallel}$, and also a lower transition field $B^*$ in thinner crystals is that nodal $p$-wave superconductivity (that has lower $T_c$, Figs. 2d, 4b) may become more energetically favourable near the surfaces due to hybridisation with topological surface states[10,50] (recall that our calculations only considered the hidden symmetry breaking and spin polarisation of bulk electronic bands). In $\beta$-PdBi$_2$, topological surface sates[28,30] have the same in-plane spin polarisation as the bulk bands[28,30]. Therefore, one can expect the overall effect of $B^{\parallel}$ to be enhanced, allowing a transition to $p$-wave superconductivity at a lower $B^*$ and suppressing the overall $\Delta(B, T)$. A detailed understanding of this effect is beyond our current work and requires further experiments using a broad range of crystal thicknesses and further development of theory.

We note that the mechanism responsible for the field-induced transition in $\beta$-PdBi$_2$ appears to be fundamentally different from the phase transitions in uranium-based heavy fermion superconductors, such as s UPt$_3$[51], UTe$_2$[3] and U$_{1-x}$Th$_x$Be$_{13}$[52]. In the latter case, superconductivity is due to f-electron pairing with large magnetic moments[53] compared to the 4 d and 6p states in $\beta$-PdBi$_2$[26]. Furthermore, the predominant theory for uranium-based superconductors is that all superconducting phases are odd-parity $p$-wave or $f$-wave with different pairing potentials ($d$-vectors), including the zero-field state, where coupling to magnetic orders determines which state is energetically favourable[2,41,54]. No magnetic orders have been detected in $\beta$-PdBi$_2$. In CeRh$_2$As$_2$, another heavy fermion system with a field-induced transition[20,21], the symmetry and spin-orbit properties are remarkably similar to $\beta$-PdBi$_2$, and the transition is also believed to be even-to-odd parity[55]. However, the presence of antiferromagnetic correlations in CeRh$_2$As$_2$[56] and the fact that it is $B^{\perp}$ (rather than $B^{\parallel}$) that is enhanced above the Pauli limit[20] implies that the underlying mechanism must be different there, too.

Finally, a finite-momentum FFLO state[57–59] would also give rise to an enhancement of $B_{c2}$ and a spatially modulated $\Delta$, resulting in normal regions which could be interpreted as nodes in tunnelling spectroscopy. However, such non-uniform superconductivity is usually energetically unfavourable and only exists for very large fields close to the Pauli limit, $B_P = 1.86$ T/K $\times T_c$. For our $\beta$-PdBi$_2$ $B^*$ and indeed $B_{c2}$ are far below $B_P \approx 5.6$ T. In transition metal dichalcogenides with strong out-of-plane Ising SOC, finite momentum pairing has been suggested at $B < B_P$[23,60]. However, such a picture does not apply to $\beta$-PdBi$_2$ where SOC is Rashba-type. In contrast, our simple model captures all features of the experimentally observed transition.

## Methods

### Crystal growth and characterisation

Single crystals of $\beta$-PdBi$_2$ were grown using a melt growth method. Pd and Bi in a molar ratio of 1:2 were sealed in an evacuated quartz tube and kept at a high temperature (1050 °C) for 6 h to ensure complete melting and mixing of the components. The temperature was then reduced to 920 °C at 50 °C/hour, the molten mixture maintained at this temperature for 24 h, then slowly cooled to 500 °C at a rate of 3 °C h$^{-1}$ and rapidly quenched into iced water. This produced cleavable single crystals, with flat surfaces as large as $\sim 6 \times 6$ mm$^2$ (Supplementary Fig. 1b). Once recovered from the quartz tube, the crystals were always handled in the argon atmosphere of a glovebox (O$_2$ < 0.1 ppm,

$H_2O < 0.1$ ppm) to prevent surface degradation. Phase purity was confirmed by X-ray diffraction ($\lambda = 1.5418$ Å, Rigaku Smartlab), see Supplementary Fig. 1b. To confirm that no phase transformations or impurities were introduced during device fabrication (see below), one of the studied devices was used for cross-sectional analysis in a scanning transmission electron microscope (STEM). To this end, we used the well-known in situ 'lift-out' method[61] and low-kV ion beam polishing[62] to prepare a thin slice of the β-PdBi$_2$ crystal removed from the active area of a SiO$_2$-PdBi$_2$-hBN-FLG stack, perpendicular to the Bi-Pd-Bi layers. The thin slice of material was then transferred to a specialist Omniprobe TEM support grid and mounted with the incident electron beam perpendicular to the plane of the lamella. The resulting STEM image shows a cross-section of the active region of the device, with a perfect arrangement of Bi and Pd atoms, see Fig. 1d.

## Magnetisation measurements

Magnetisation measurements were carried out using a commercial SQUID magnetometer MPMS XL7 (Quantum Design). Samples for these measurements (that we refer to as 'bulk') were cleaved off the same melt-grown β-PdBi$_2$ crystal as the one used to fabricate tunnel junctions. Typical sample dimensions for magnetisation measurements were $(0.01–0.1) \times 4 \times 4$ mm. Prior to being placed in the magnetometer, samples were mounted inside low-magnetic background plastic straws, taking care to protect them from exposure to air. In the zero-field-cooling (ZFC) mode of DC measurements, the sample was first cooled down to the lowest available temperature (1.8 K) in zero magnetic field, then a finite field $B$ applied and magnetisation $M(B)$ measured as a function of an increasing temperature $T$. In field-cooling (FC) mode, a field $B$ was applied above $T_c$ (typically at 10–15 K), and magnetisation was measured as a function of decreasing $T$. The superconducting fraction was found as $f = (1 - N)\, 4\pi |dM/dH|/V$, where $N$ is the demagnetisation factor and $V$ the sample's volume. This yielded $f = 1$, i.e., all our crystals were 100% superconducting.

The superconducting coherence length $\xi$ and magnetic field penetration depth $\lambda$ were found from the measured critical fields $B_{c1}$ and $B_{c2}$ using the standard expressions[63] $B_{c2} = \Phi_0/2\pi\xi^2$ and $B_{c1} = \left(\Phi_0/4\pi\lambda^2\right)[\ln k + \alpha(k)]$, where $\alpha(k) = 0.5 + (1 + \ln 2)/(2k - \sqrt{2} + 2)$. The measured critical fields (Supplementary Fig. 1a) were accurately reproducible for all studied crystals and did not depend on the crystal thickness in this 'bulk' limit (thickness $d$ between 10 and 100 μm). At the lowest measurement temperature $T = 1.8$ K, we found $B_{c1}(1.8K) = 7$ mT, $B_{c2}^{\parallel}(1.8K) = 0.68$ T, $B_{c2}^{\perp}(1.8K) = 0.56$ T. The upper critical field for these crystals, $B_{c2}(T)$, is accurately described by the standard WHH theory[64], see inset in Supplementary Fig. 1a, yielding extrapolated values $B_{c2}^{\parallel}(0) \approx 0.9$ T and $B_{c2}^{\perp}(0) \approx 0.74$ T and a corresponding in-plane coherence length $\xi_{ab}(0) \approx 22$ nm. Low-$T$ penetration depth was estimated using $B_{c1}(0) \approx 9$ mT, $\lambda(0) \approx 240$ nm.

## Device fabrication

The layered nature of β-PdBi$_2$ allows it to be exfoliated similarly to graphite and stacked with other van der Waals materials. To build planar SIN tunnel junctions, we used 50–140 nm thick PdBi$_2$ crystals as the superconducting electrode, 2-3 layer thick hBN as an atomically flat tunnel barrier and few-layer graphene (FLG) as the normal metal, see schematic and an image of a typical device in Fig. 1b, c. To this end, we used a dry transfer 'stamping' method where PdBi$_2$, FLG and hBN were exfoliated individually onto Si/SiO$_x$ wafers. As the first step, Polypropylene carbonate (PPC) was spin-coated on polydimethylsiloxane (PDMS) mounted on a glass slide. This assembly was then used to pick up ~ 25 nm thick top encapsulating layer of hBN (see ref. 65). This hBN crystal was then used to lift FLG strips from the Si/SiO$_x$ substrate, followed by picking up of a 2–3 layer-thick hBN flake from the thicker hBN crystal deposited initially on the Si/SiO$_x$ substrate (this served as the tunnel barrier). Finally, the assembled stack was deposited onto a suitable PdBi$_2$ flake by detaching it from the PDMS/PPC stamp. Exfoliation of PdBi$_2$ and the final stacking

step were carried out in the protective atmosphere of an Ar-filled glovebox, to avoid degradation in air. Finally, Cr/Au contacts to FLG or directly to the PdBi$_2$ crystal (Fig. 1b) were patterned using electron beam lithography: the encapsulating hBN layer over the contact areas was removed using reactive ion plasma etching and Cr/Au contacts deposited by thermal evaporation. To ascertain that the fabrication procedure did not affect the quality and crystallinity of PdBi$_2$, we used cross-sectional transmission microscopy as described in the 'Crystal growth and characterisation' section.

As explained in the main text, we studied a relatively narrow range of thicknesses of PdBi$_2$ crystals in our tunnelling devices, with 50 nm being the thinnest. It would be interesting to also study much thinner, atomically thin, β-PdBi$_2$ but mechanical exfoliation of this material is, unfortunately, difficult. Our many attempts to produce crystals thinner than 50 nm were unsuccessful, as each successive exfoliation step made the crystals thinner but also smaller, with lateral dimensions quickly becoming less than a couple of microns and so unsuitable for making a device.

## Tight binding Hamiltonian

Electronic states at the Fermi surface of β-PdBi$_2$ originate primarily from Bi p-orbitals and exhibit spin-momentum locking due to the strong atomic spin-orbit coupling (SOC)[26,28]. Accordingly, we model β-PdBi$_2$ as a stack of Bi bilayers and construct a minimal tight-binding model from Bi p-orbitals (details of the model in Supplementary Note 2.2). We focus on the bulk bands with clear 2D character, i.e., those that generate Fermi surfaces that are nearly flat in the direction orthogonal to PdBi planes and originate from (predominantly) in-plane Bi orbitals that are only weakly hybridised with those of neighbouring bilayers. Due to the globally centrosymmetric atomic arrangement in 3D crystals of β-PdBi$_2$, Bi bands exhibit a twofold sublayer degeneracy. However, electrons at Bi sites experience a locally non-centrosymmetric environment[44]. The local crystal field couples in-plane and out-of-plane Bi p-orbitals and, in combination with the atomic SOC, $\lambda L \cdot S$, gives rise to a Rashba-like SOC[44] and in-plane spin-momentum locking at the Fermi surface. Since Bi atoms of different sublayers are related by inversion symmetry (Supplementary Fig. 7), their electrons experience opposite crystal fields and Rashba-like SOCs. Therefore, the twofold degenerate bands exhibit opposite in-plane helical spin polarisations (see the sketch in Fig. 4a). As briefly discussed in the main text, it is the spin helicity of the bulk bands at the Fermi surface that plays a major role in the response of β-PdBi$_2$ superconductivity to magnetic fields. To study this effect in detail with semi-analytical techniques, we have derived a continuum model by expanding the 2D-like Bi band structure at the Fermi surface up to second order in wavevector $\boldsymbol{k}$ around the Γ point. To simplify the problem as much as possible, we ignore the warping terms that give rise to a square-like Fermi surface, as well as the $\boldsymbol{k}$-dependent interlayer hopping terms. This results in a circularly symmetric Fermi surface. Including a Zeeman term due to the magnetic field, the resulting continuum Hamiltonian reads

$$H_0 = \frac{k^2}{2m} - \mu - \epsilon\sigma_x + \alpha\sigma_z\left(k_x s_y - k_y s_x\right) - h s_x, \qquad (7)$$

where $\alpha$ is the Rashba spin-orbit strength, $m$ the effective mass, $\mu$ the chemical potential, $\epsilon$ the hopping parameter between Bi sublayers and $h$ the Zeeman energy. In this equation, $s_i$ and $\sigma_i$ ($i = x, y, z$) are Pauli matrices operating on the spin and sublayer degrees of freedom, respectively. We use $\alpha = 0.81$ eV, $m = -0.43$ eV$^{-1}$ and $\epsilon = 0.63$ eV, which reproduce the bands around the Γ point well. The typical value of the chemical potential is $\mu = -2.22$ eV. Next, we consider the effect of the following interaction Hamiltonian:

$$H_{int} = -U\left(n_1^2 + n_2^2\right) - 2V n_1 n_2, \qquad (8)$$

where $n_i$ is the electron density in the Bi sublayer $i$, and $U$ and $V$ are local (Hubbard-like) and interlayer density-density interactions, respectively. While $U$ allows only s-wave (spin-singlet) pairing, $V$ enables also spin-polarised pairing since paired electrons on different layers can have aligned spins. We apply a mean-field decomposition to $H_{int}$ into two candidate pairing channels by introducing an s-wave order parameter $\psi$ and a spin-polarised order parameter $\eta$. Both $\psi$ and $\eta$ pair electrons with opposite momenta; electrons paired by $\eta$ have parallel spins. The calculated free energies for s-wave and triplet states are presented and discussed in the main text.

### Fitting tunnelling data

In zero field, the suppression of PdBi$_2$ superconductivity with increasing $T$ shows standard BCS behaviour, with coherence peaks at $eV_b \approx \pm \Delta$, as seen in the $G(V_b, T)$ in Fig. 1f. To extract the gap values, $\Delta(T)$, from measured individual spectra, these were first normalised by dividing by $G(V_b, T>T_c)$, i.e., by the spectra measured in the same range of $V_b$ above $T_c$. We then calculated the DoS using Eq. (2) in the main text and numerically integrated with the Fermi-Dirac derivative, Eq. (1), for a given set of trial parameters $(\Delta, \Gamma)$. This procedure was repeated iteratively to find the parameters that minimised the sum of square residuals between the model and the data, and to extract $\Delta(T)$ shown in Fig. 1e.

To analyse the dependence of the measured spectra on $B$, we followed the theory developed by Maki[37], which itself used the theoretical framework formulated earlier by Abrikosov and Gor'kov[66] and Skalski et al.[67] for the DoS of a superconductor with magnetic impurities. A unifying concept in these theories is a time-reversal breaking perturbation caused by either the applied field or magnetic impurities[43]. Early tunnelling experiments on thin films of conventional s-wave superconductors in parallel magnetic field[38,39] showed that the field modifies not only the energy gap but also the functional form of the DoS, in excellent agreement with the Maki theory. In this scenario, the superconducting DoS is $N(E) = N_\uparrow(E) + N_\downarrow(E)$, where $N_{\uparrow\downarrow}$ is the DoS for each spin species given by

$$\frac{N_{\uparrow\downarrow}(E)}{N_N(0)} = \frac{1}{2}\operatorname{sgn}(E)\operatorname{Re}\left(\frac{u_\pm}{\sqrt{u_\pm^2 - 1}}\right), \qquad (9)$$

where the parameter $u_+$ ($u_-$) corresponds to spin up (spin down) and must be determined self-consistently from the equation

$$u_\pm = \frac{E \mp \mu B}{\Delta} + \frac{\zeta_\pm}{\sqrt{1 - u_\pm^2}} \qquad (10)$$

Here $\mu B$ is the Zeeman energy, and $\zeta$ parametrises the orbital pair breaking due to the breaking of time-reversal symmetry. To calculate the DoS for a given set of parameters $(\Delta, B, \zeta)$, we numerically solved Eqs. (9–10) at each $E$ to give $N(E)$. The conductance is then given by Eq. (1) in the main text, similar to the zero-field case. The g-factor of β-PdBi$_2$ is $\approx 2$ and the externally applied field penetrates our ~100 nm thick crystals uniformly, so we can set $\mu = \mu_B$ (Bohr magneton) and take $B$ in (10) equal to the applied field $B^\parallel$. We are then left with only two fitting parameters, $(\Delta, \zeta)$; their field dependences $\Delta(B)$ and $\zeta(B)$ can be found by solving simultaneously the transcendental equations[39]

$$\zeta = \frac{1}{2}\left(\frac{\Delta_0}{\Delta}\right)\left(\frac{B}{B_{c2}}\right)^2 \qquad (11)$$

$$\ln\left(\frac{\Delta}{\Delta_0}\right) = \begin{cases} -\frac{1}{4}\pi\zeta, & \zeta \le 1 \\ -\cosh^{-1}(\zeta) - \frac{1}{2}\left[\zeta\sin^{-1}\left(\frac{1}{\zeta}\right) - \sqrt{1 - \frac{1}{\zeta^2}}\right], & \zeta > 1 \end{cases} \qquad (12)$$

where $\Delta_0 \equiv \Delta(B = 0)$. The solution for $\zeta(B)$ shows that ZBC is only non-zero when $\zeta > 1$ which occurs at $B/B_{c2} > \sqrt{2}e^{-\frac{\pi}{8}} \approx 95\%$[39,43,67]. In real s-wave superconductors, the value of $\zeta$ can be higher for equivalent $B$ (green dashed line in Fig. 3c) due to finite mean-free path effects[38,39], but ZBC is still practically zero until $B/B_{c2} \approx 80\%$ as shown by the green dashed curve in Fig. 3d, in agreement with more detailed calculations by Strassler and Wyder[68]. This is in clear contrast to our findings, where ZBC increases almost immediately after the field reaches $B \sim B^*$, see Figs. 2c and 3d.

Let us note that the full Maki theory[37] contains one more term in Eq. (10), taking into account the effect of spin-orbit scattering. However, this term only needs to be included for large Zeeman splitting, such that $\mu_B B \gg \Delta$ or $B > 10$T[69] and is therefore not relevant for β-PdBi$_2$.

In the low-field regime $B < B^*$ the Maki theory provides accurate fits to our experimental spectra and $\zeta(B)$ almost exactly follows the form predicted by Eqs. (11), (12) if we set $B_c^{s-wave} = 0.25$ T, see Supplementary Fig. 5 for details. On the other hand, attempting to apply the Maki theory to the 'V'-shaped spectra at $B > B^*$ results in poor fits even if $\zeta(B)$ is treated as a fitting parameter and allowed to take on unphysically large values compared to theory expectations for the corresponding range of $B/B_{c2}$, see Supplementary Fig. 5b (here $B_{c2} \approx 1.6$T is the actual experimentally measured critical field). Using the theoretically predicted values of $\zeta(B)$ at $B > B^*$ (green dashed line in Fig. 3c) results in large discrepancies between the expected and observed spectra.

In contrast, nodal DoS (Supplementary Note 2.1) fits the data at $B > B^*$ well, see Fig. 3b and Supplementary Fig. 5b. Here, we use a known approach to analysing tunnelling spectra of unconventional superconductors by incorporating all pair-breaking effects into a field-dependent imaginary part of the self-energy, $\Gamma$, by analogy with the zero-field Dyne's model, and extract $\Delta(B)$ and $\Gamma(B)$ for each spectrum above $B^*$ using Eq. (3). As illustrated in Supplementary Fig. 3d, e, the effect of $\Gamma$ in this case is to increase the ZBC already at low $B$, unlike the result of the s-wave Maki theory. This agrees qualitatively with calculations for specific pair-breaking perturbations in e.g., heavy fermion superconductors[70].

## Data availability

The authors declare that the data supporting the findings of this study are available within the paper and its Supplementary Information/Source Data files. Source data are provided in this paper.

## Code availability

The software code used in this work is available as a supplementary information file.

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

## Acknowledgements

We acknowledge financial support from Horizon 2020 Graphene Flagship Project (Core 3), I.V.G., A.K.G., J.B. and M.K.; the Lloyd's Register Foundation, A.K.G. and W.K.; the Engineering and Physical Sciences Research Council (EPSRC; Grant Nos EP/V007033/1 and EP/Z531121/1), S.H. and A.K.G.; A.P. acknowledges support from the Leverhulme Trust under the Grant Agreement No. RPG-2019-363; A.P. and I.V.G acknowledge support from the EU Horizon 2020 MSCA-RISE-2019 programme (Project No. 873028 HYDROTRONICS); L.P., G.H.P. and W.K. acknowledge support from the EPSRC CDT in Science and Applications of Graphene and Related Nanomaterials (EPSRC Grant EP/L01548X/1).

## Author contributions

I.V.G and W.K. initiated, and I.V.G. and L.P. led the project. G.H.P., A.P. and N.W. developed theoretical models and carried out theoretical analysis. R.J. and J.B. fabricated the devices with a contribution from S.K.; W.K. and G.B. grew PbBi$_2$ crystals with help from I.T. Crystal structures were imaged by Y.Z. and S.H.; L.P. and W.K. carried out tunnelling spectroscopy and transport measurements with help of M.K.; W.K. and Z.J. carried out magnetisation measurements. L.P. and I.V.G. analysed the data with contributions from Z.J. and W.K.; I.V.G., L.P., G.H.P. and A.P. wrote the manuscript with contributions from A.K.G. All authors contributed to discussions.

## Competing interests

The authors declare no competing interests.
