## [Transparent Peer Review file · Nature Communications]

Multiphase superconductivity in PdBi₂

Corresponding Author: Professor Irina Grigorieva

Version 0:

Reviewer comments:

Reviewer #1

(Remarks to the Author)

Powell et al. studied the superconducting transition in thin PbBi₂ films using four-terminal resistance measurements and tunneling spectroscopy. The authors carefully examined the magnetic field and temperature dependence, arguing a phase transition from s-wave pairing to p-wave pairing induced by an in-plane magnetic field. The main evidence presented is a kink in the superconducting gap at B* and a change in the shape of the spectrum. The authors also constructed a tight-binding model to explain the observed phase transition. P-wave superconductivity is of great interest to the community due to its potential applications in quantum computing. However, the evidence provided here is too weak to support such a significant claim. I suggest the authors tone down their conclusions. Below, I list my comments:

1, The kink in Fig. 2a could be caused by trapped flux. With a thickness of about 100 nm, flux can be created even in the in-plane field geometry. Does the author observe hysteresis when sweeping the magnetic field across B*? Does the value of B* vary with the azimuthal angle of the magnetic field? Does the value of B* depend on the thickness of the sample?

2, The authors showed that T_c decreases with sample thickness. I suggest they fabricate and measure samples with the thinnest possible thickness to eliminate orbital effects.

3, The procedure to extract B_{c2} in Fig. 2d needs clarification. The B_{c2} following two trends is quite interesting.

4, While I appreciate the careful fitting and study of the spectra, this fitting has many parameters and only works well in a limited range. Using the shape and fitted parameters as the main argument for p-wave pairing is not solid. More direct evidence is needed.

Overall, the study presents interesting data, but the conclusions about p-wave superconductivity require stronger evidence. I recommend significant revisions to address the major comments and provide additional evidence or analysis to support their claims.

Reviewer #2

(Remarks to the Author)

This manuscript reports the characterization of the superconducting phase in PdBi₂ samples with a thickness of 50-80 nm. Through the fabrication of tunnel junctions and measurement of the tunneling conductance, a threshold in-plane magnetic field, B* ~ 0.2 T, was identified. This threshold delineates two distinct regimes, each with different rates of field-induced weakening of the superconducting gap. Notably, this behavior is absent under out-of-plane magnetic fields. Additionally, an anomaly was detected in the B-T phase diagram using resistance data, occurring at 2-3 times B*. By analyzing the tunneling conductance data, the authors suggest that below B*, the superconducting pairing is of the s-wave type, whereas above B*, it transitions to p-wave pairing. This hypothesis is further supported by a tight-binding model that highlights the local inversion symmetry breaking, which can lead to a hidden Rashba-like splitting of the electronic bands.

Overall, I found the reported results highly interesting. If confirmed, they would represent a significant advancement in the study of unconventional superconductivity driven by the interplay between spin-orbit coupling and magnetic fields. Before recommending the publication of this work, I hope the authors can address the following questions.

1. If I understand the results correctly, the detection of B* depends on the sample thickness. However, the specific thickness range for the reported phenomena seems to be missing in the current manuscript. What occurs in samples thinner than 50 nm? Have the authors measured such samples?

2. A question related to the first one: the authors argue that thin samples can host p-wave superconductivity when the

surface layers hybridize with topological surface states. In this scenario, what is the estimated thickness of the surface layer hosting p-wave superconductivity? For samples thicker than this estimated thickness, will there be a mixture of p-wave and s-wave layers?

3. I appreciate the authors' effort to ensure high sample quality by using STEM for verification after characterizing superconductivity. This method selectively probes a small portion of the flake. I notice that some features in the resistance data are puzzling. In Figure 2e and Extended Data Figure 2b, there are kinks in the field dependence of the resistance, particularly near 3 K at low field. How can sample inhomogeneity be ruled out as the cause of these features?

4. How can the absence of the vortex effect under in-plane field be justified? It was mentioned that $d \sim 2\xi \ll \lambda$, but the values for ξ and λ were not specified.

5. It was stated in the abstract and the text that the gap closes and reopens. This description implies that the gap completely closes at some field value, which contradicts what is shown in Figure 3. However, the complete closing of the s-wave gap is necessary for the field-induced phase transition.

6. Even for an s-wave superconducting with the reported thickness and T_c , the value of $B^* = 0.2T$ for closing the gap seems quite low. The authors did not discuss the fitted depairing strength according to Maki's theory. Is it compatible with the parameters for PdBi2?

Version 1:

Reviewer comments:

Reviewer #1

(Remarks to the Author)

The revised manuscript and response address all my concerns. I recommend the publication of the paper.

Reviewer #2

(Remarks to the Author)

The authors have well addressed my questions. I have no further comments and recommend the manuscript for publication.

Reviewer #1 (Remarks to the Author):

Powell et al. studied the superconducting transition in thin PbBi₂ films using four-terminal resistance measurements and tunneling spectroscopy. The authors carefully examined the magnetic field and temperature dependence, arguing a phase transition from s-wave pairing to p-wave pairing induced by an in-plane magnetic field. The main evidence presented is a kink in the superconducting gap at B and a change in the shape of the spectrum. The authors also constructed a tight-binding model to explain the observed phase transition. P-wave superconductivity is of great interest to the community due to its potential applications in quantum computing.*

We are grateful to the Reviewer for highlighting the great interest in the community in the subject of our study.

However, the evidence provided here is too weak to support such a significant claim. I suggest the authors tone down their conclusions.

We fully agree with the Reviewer that an unequivocal claim of p-wave superconductivity in PbBi₂ would be premature and further experimental and theoretical evidence is certainly required. This is a challenging task for any single work, which probably requires to develop a consensus, as clearly demonstrated by decades-long debates and still puzzling unconventional superconductivity of SrRuO₄. Rather than definitively claiming p-wave superconductivity, the emphasis in our work is on the magnetic-field-induced transition between two different types of superconducting behaviour, seen as different modes of the evolution of its superconducting order parameter and distinct regions in the phase diagram. We interpret this as a transition from conventional s-wave pairing to nodal pairing whose experimental manifestation is consistent with p-wave order parameter. The tunnelling spectra, in particular, provide evidence of line nodes in the gap above the transition field and – indirectly – of much enhanced pair-breaking effects. At the same time our theoretical analysis of the effect of magnetic field on β -PdBi₂ superconductivity, with its specific crystal and electronic structure, shows that p-wave pairing becomes energetically favourable above a certain applied field if it is parallel to the crystal's ab plane. Putting the two together allows us to argue that the observed field-induced transition is consistent with a transition from s-wave to nodal p-wave pairing.

We are sorry if the wording in the original manuscript implied a stronger claim, and we changed the relevant statements in the revised manuscript to tone it down, as suggested by the Reviewer. Furthermore, we have expanded the discussion by considering possible alternative types of nodal pairing and showing that these would be inconsistent with our model and/or experimental observations. As discussed in Supplementary Notes, the states of a cylindrical Fermi surface for a bilayer Rashba system can be described by a continuum Hamiltonian. We then assume that two competing interactions exist in this system: a local (Hubbard) one, and a non-local one that couples electrons sitting on nearest-neighbouring sites that belong to different layers. This is the simplest model that reproduces the main features of β -PdBi₂ band structure, including the nontrivial spin helicity of bulk and surface states, and it predicts a phase transition in a sufficiently strong in-plane field. The only allowed pairing channels in such a system are nodeless s-wave states, an odd-parity pair density wave (PDW) state that changes sign each sublayer, and $|m_L|=0$ and $|m_L|=1$ spin-triplet states [1]. While the PDW and $|m_L|=0$ triplet phases have point nodes at $k_x = k_y = 0$ and are therefore not compatible with the experimental spectra [2], the only ones that host nodal lines are components of the $|m_L|=1$ triplet pairing. This effectively p-wave triplet phase is therefore the simplest nodal phase that can exist in this system. We note that higher symmetry d-wave states can also have nodal lines, but they are not compatible with the type of interaction assumed in our simple model. One would have to consider more contrived non-local interactions to stabilise such phases.

We also emphasise the significance of an in-plane magnetic field. It can induce a substantial modification of the band structure of β -PdBi₂ because of the spin-momentum locking due to the strong spin-orbit coupling. In turn, these changes in the band structure make the effectively p-wave triplet phase more stable than the s-wave above the transition field, a result that is compatible with experimental observations. This dependence on the applied magnetic field also suggests that a singlet d-wave pairing may not be suitable to describe the observed transition, because a Zeeman field, that only affects the spin, would not distinguish between two singlet states, or favour one over the other.

[1] Nakosai, S., Tanaka, Y. & Nagaosa, N. Topological Superconductivity in Bilayer Rashba System. *Phys. Rev. Lett.* **108**, 147003 (2012).

[2] Sigrist, M. & Ueda, K. Phenomenological theory of unconventional superconductivity. *Rev. Mod. Phys.* **63**, 239–311 (1991).

Below, I list my comments:

1, The kink in Fig. 2a could be caused by trapped flux. With a thickness of about 100 nm, flux can be created even in the in-plane field geometry. Does the author observe hysteresis when sweeping the magnetic field across B? Does the value of B* vary with the azimuthal angle of the magnetic field? Does the value of B* depend on the thickness of the sample?*

The thickness of all PdBi₂ crystals in our devices is 2 to 4 times smaller than the magnetic field penetration depth for this material, ~ 240 nm at low temperatures (the latter has been evaluated from our measurements of the lower critical field). This implies that in all our measurements the in-plane magnetic field penetrates the crystals uniformly and the notion of trapped flux does not apply. Indeed, with experimental accuracy we did not notice any difference in the tunnelling spectra measured in an increasing/decreasing field. In terms of the angular dependence, we agree with the Reviewer that B* can be expected to vary with the azimuthal angle as this would correspond to a decrease in the in-plane component of the field. Unfortunately, our experimental setup only allowed measurements in either in-plane or out-plane field configuration. We agree that a detailed study of the angular dependence would provide further insight into the effect of the magnetic field, and this will be subject of future work.

The value of B* does appear to depend on the thickness of PdBi₂ crystals – Fig. 3a shows data for 80 nm and 50 nm, with B* of ~ 0.2 T and ~ 0.1 T respectively. For the devices of similar thickness we found similar values of B* (two of our devices had ~ 75 nm and 80 nm thick PdBi₂). We speculate that the observed thickness dependence of B* may be related to a greater role of the topological surface states in thinner samples. This effect would be additional to hidden symmetry breaking and spin polarisation of the bulk electronic bands considered in our tight-binding calculations. Given similar spin polarization of the topological surface states established in the literature (refs. [29,30] in the manuscript) it is reasonable to expect that hybridization with these states may shift the transition to nodal pairing to a lower magnetic field and decrease the value of B*. See also our reply to point 2 below.

We apologise that the above aspects of our work were not explained sufficiently clearly. In the revised manuscript we have added details of experimental determination of the superconducting characteristics of our PdBi₂ crystals (penetration depth, coherence length) and of the sample-to-sample variation of B*.

2, The authors showed that Tc decreases with sample thickness. I suggest they fabricate and measure samples with the thinnest possible thickness to eliminate orbital effects.

We appreciate the Reviewer's suggestion to study thinner crystals and agree it would be interesting. Unfortunately, mechanical exfoliation of this material is difficult. Our many attempts to produce crystals thinner than 50 nm were unsuccessful as each successive exfoliation step makes the crystals thinner but also smaller, with lateral dimensions quickly becoming less than a couple of microns and so unsuitable for making a device. We spent 3 years on the project trying to do this. Following the Reviewer's comment, we gave it another go, which again resulted in suitable (>a few microns in size) crystals only thicker than 80 nm. Realistically, we don't see a way to achieve thinner devices.

We have explained this in the revised manuscript.

3, The procedure to extract B_{c2} in Fig. 2d needs clarification. The B_{c2} following two trends is quite interesting.

We apologise for not explaining this clearly. B_{c2} values shown in Fig. 2d were extracted from the resistance vs magnetic field, $R(B)$, curves measured at different temperatures (such as shown in Fig. 2e). B_{c2} in Fig. 2d is taken as the magnetic field at which the resistance is 90% of the normal-state value. We also tried different criteria and obtained the same phase diagram, which is consistent with the sharpness of the resistance curves.

4, While I appreciate the careful fitting and study of the spectra, this fitting has many parameters and only works well in a limited range. Using the shape and fitted parameters as the main argument for p-wave pairing is not solid. More direct evidence is needed.

There are only two fitting parameters for each of the two models that we used to analyse the tunnelling spectra and we use both qualitative and quantitative arguments (spectral shape and parameter values extracted from fitting) to claim that the pairing becomes unconventional at fields $B > B^*$. The Reviewer is right that the Maki theory (known to provide an accurate description of the effect of magnetic field for s-wave superconductors up to B_{c2}) describes our data well only at fields $B \leq B^*$ and the agreement breaks down at higher fields. This implies a change in the type of pairing at $B > B^*$. As to the actual symmetry of the order parameter in higher fields, we believe that analysis of the tunnelling spectra allows us to say that it must be nodal (with line nodes) but in principle can have symmetry other than p-wave. On the basis of experiment alone p-wave must be treated as an assumption. However, our theoretical analysis of the effect of magnetic field on the specific band structure of PdBi_2 identifies p-wave pairing as the most likely candidate for sufficiently high magnetic fields.

To address the Reviewer's concern, the discussion in the revised manuscript is more nuanced and especially the conclusions refer to evidence for a transition to nodal pairing, with p-wave being not a definitive but most likely candidate.

We have also expanded the description of our fitting procedure and moved parts of it from Methods to the main text. We now emphasise that the parameters needed to accurately describe the experimental spectra are only Δ (order parameter) and ζ (depairing strength) for s-wave pairing (Maki model), and Δ and a generalised 'broadening' parameter Γ for the nodal pairing. Other quantities appearing in eqs. (10)-(12) are fixed: T is the experimental temperature ($0.3\text{K}=0.1T_c$), and $g \approx 2$ fixes the Zeeman energy $\mu_B B$. In the full Maki theory (refs. 34,40 in the manuscript) there is also an additional term in eq. (10) containing spin-orbit scattering strength b . However, it is significant only for $\mu_B B \gg \Delta$ (Meservey et al, Phys. Rev. B 11, 4224, 1975) or magnetic fields $B > 10T$, much larger than B_{c2} for PdBi_2 , so can be set to zero in our case. We have explained this in the revised version of the manuscript.

Overall, the study presents interesting data, but the conclusions about p-wave superconductivity require stronger evidence. I recommend significant revisions to address the major comments and provide additional evidence or analysis to support their claims.

We are grateful to the Reviewer for describing our data as interesting. The revised manuscript includes additional analysis of how the parameters describing the density of states for different types of superconducting pairing affect the tunnelling spectra (revised Supplementary Figure 5 and associated discussion) and a discussion of possible alternative types of superconducting pairing. We also toned down the claim of p-wave superconductivity and made the discussion more nuanced as described above. Overall, following the Reviewer's comments, the revised manuscript emphasizes that the invoked p-type superconductivity in PdBi, being a likely scenario, requires further confirmation. We hope that this will stimulate theorists to think about alternative models while experimentalists could apply alternative probing techniques in search for definite answers.

Reviewer #2 (Remarks to the Author):

This manuscript reports the characterization of the superconducting phase in PdBi₂ samples with a thickness of 50-80 nm. Through the fabrication of tunnel junctions and measurement of the tunneling conductance, a threshold in-plane magnetic field, $B^ \sim 0.2$ T, was identified. This threshold delineates two distinct regimes, each with different rates of field-induced weakening of the superconducting gap. Notably, this behavior is absent under out-of-plane magnetic fields. Additionally, an anomaly was detected in the B-T phase diagram using resistance data, occurring at 2-3 times B^* . By analyzing the tunneling conductance data, the authors suggest that below B^* , the superconducting pairing is of the s-wave type, whereas above B^* , it transitions to p-wave pairing. This hypothesis is further supported by a tight-binding model that highlights the local inversion symmetry breaking, which can lead to a hidden Rashba-like splitting of the electronic bands.*

Overall, I found the reported results highly interesting. If confirmed, they would represent a significant advancement in the study of unconventional superconductivity driven by the interplay between spin-orbit coupling and magnetic fields.

We are grateful for this positive assessment of our results.

Before recommending the publication of this work, I hope the authors can address the following questions.

1. If I understand the results correctly, the detection of B^ depends on the sample thickness. However, the specific thickness range for the reported phenomena seems to be missing in the current manuscript. What occurs in samples thinner than 50 nm? Have the authors measured such samples?*

Our tunnelling devices contained β -PdBi₂ crystals of thicknesses between 50 and 140 nm. We have made many attempts to exfoliate thinner crystals but unfortunately it proved to be impossible. Despite its layered structure, exfoliation of this material is difficult. Each successive exfoliation step makes the crystals thinner but also smaller, with lateral dimensions quickly becoming less than a couple of microns and so unsuitable for device fabrication. Following the Reviewer's comment, we gave it another go, which again resulted in suitable (>a few microns in size) crystals only thicker than 80 nm. Realistically, we don't see a way to achieve thinner devices. We have explained this in the revised manuscript.

Within the range of thicknesses that we were able to achieve, B^* does appear to depend on the thickness (it is found to be lower for our 50nm, while for the two devices with a similar thickness, ~ 80

nm, B^* values are similar). Still, we do not have enough statistics to make a substantiated claim about this and we clarified this in the revised manuscript.

2. A question related to the first one: the authors argue that thin samples can host p-wave superconductivity when the surface layers hybridize with topological surface states. In this scenario, what is the estimated thickness of the surface layer hosting p-wave superconductivity? For samples thicker than this estimated thickness, will there be a mixture of p-wave and s-wave layers?

We are sorry for not being sufficiently clear. Our theoretical analysis is based on bulk electronic bands of β -PdBi₂: as explained in ‘Tight binding Hamiltonian’ section in Methods and in Supplementary Information, the description of the superconducting state of β -PdBi₂ and its response to the magnetic field is based on the spin-polarised bulk bands; the presence of the topological surface states is not included in the theory. This implies that the transition to triplet pairing can be expected in bulk crystals. That said, the Reviewer is right that hybridization with topological surface states is likely to play a role, too (e.g. [Kuang et al, Adv. Mater. 33, 2103257 (2021); Kashiwaya et al, Phys. Rev. B. 53, 2667 (1996)], creating favourable conditions for the transition to p-wave pairing at a lower B . This effect should be particularly noticeable for thin crystals of β -PdBi₂ and may explain the thickness dependence of the critical parameters of superconductivity reported in the literature (T_c , B_{c2}) and also seen in our study, as well as our observation of a lower B^* for our thinnest (50 nm) device.

Beyond this, the narrow range of crystal thicknesses in our experiment and the employed experimental technique (tunnelling spectroscopy) do not allow us to unambiguously determine whether a phase separation into s-wave and p-wave layers does occur in truly bulk crystals and what is the critical thickness for this to happen. On the scale of the characteristic superconducting lengths (λ , ξ) PdBi₂ crystals in all our devices are ‘thin’, $d \sim (2-3)\xi \ll \lambda$ and hybridization between different electronic states contributing to superconductivity is likely throughout the crystal’s bulk [Kuang et al, Adv. Mater. 33, 2103257 (2021), Kashiwaya et al, Phys. Rev. B. 53, 2667 (1996)]. Additionally, contributions to tunnelling conductance detected by planar tunnelling spectroscopy decay with the distance from the surface and are likely to be limited to a few ξ [Wolf, Principles of electron tunnelling spectroscopy, Oxford Univ. Press, 2012].

We have clarified these points in the discussion in the revised manuscript and hope the Reviewer will agree that answering the question about possible phase separation and its critical thickness is beyond a single study.

3. I appreciate the authors' effort to ensure high sample quality by using STEM for verification after characterizing superconductivity. This method selectively probes a small portion of the flake. I notice that some features in the resistance data are puzzling. In Figure 2e and Extended Data Figure 2b, there are kinks in the field dependence of the resistance, particularly near 3 K at low field. How can sample inhomogeneity be ruled out as the cause of these features?

We agree with the Reviewer that STEM is only looking at a small portion of the crystal. Our main reason to use STEM was to verify that our fabrication procedure did not cause phase transformations or introduced defects in the PdBi₂ crystals. That said, the image shown in Fig. 1d is representative of many more images obtained in the STEM experiment (which were practically identical). Other indications of the high crystal quality are the sharp, single-phase XRD peaks (Extended Data Fig. 1b) and very low/vanishing pinning as seen as reversible magnetization shown in Extended Data Figure 1 (now Supplementary Figure 1). Additionally, we have now done magnetoresistance measurements on

relatively thick ($\sim 1 \mu\text{m}$ thick) slabs exfoliated from the 'parent' bulk crystal and extracted the mean free path $\sim 25\text{nm}$. These data have now been added to the revised manuscript.

We are grateful to the Reviewer for noticing an irregularity in Fig. 2e. The curve in question corresponds to a temperature less than 0.1K below T_c , where even the smallest non-uniformities in the studied crystals are strongly emphasised. This can explain the observed kink. We now realise that including this curve was confusing and in the revised manuscript it has been removed from the figure.

As to the resistance curves shown in Extended Data Fig. 2b, the kinks at low temperatures and relatively broad transitions are indicative of the presence of vortices, as expected, as these measurements are done in out-of-plane field. In the revised manuscript we have corrected Fig. 2e and explained the difference between Fig. 2e and Extended Data Fig. 2b (now Supplementary Figure 2b) in more detail.

4. How can the absence of the vortex effect under in-plane field be justified? It was mentioned that $d \sim 2\xi \ll \lambda$, but the values for ξ and λ were not specified.

We are sorry for this omission. In the original manuscript we specified the values of ξ and λ only in Methods (as determined from magnetization measurements). These are now quoted in the main text. The thickness of all PdBi_2 crystals in our devices is 2 to 4 times smaller than the magnetic field penetration depth for this material, $\sim 240 \text{ nm}$ at low temperatures. This implies that in all our measurements the in-plane magnetic field penetrates the crystals uniformly. We cannot completely rule out the presence of one or two vortex cores in the device but, as our measurements in the out-of-plane field show, even the presence of a dense array of vortices does not produce any of the effects observed for in-plane fields (V-shaped spectra, rapid increase in the zero-bias conductance at and above B^*).

5. It was stated in the abstract and the text that the gap closes and reopens. This description implies that the gap completely closes at some field value, which contradicts what is shown in Figure 3. However, the complete closing of the s-wave gap is necessary for the field-induced phase transition.

We are sorry for the poor description and now realise that it is indeed misleading. We do observe a pronounced suppression (rather than closing) of the s-wave superconducting gap at low fields (below B^*) which is then followed by the appearance of a new order parameter (rather than reopening of a gap) indicated by the changes in the spectral shape and other characteristics of the tunnelling spectra. A suppression (but not closing) of the s-wave gap is consistent with the appearance of nuclei of a new phase (corresponding to the suggested p-wave pairing) at B^* as expected for a first-order - rather than second-order - phase transition, in agreement with our theory. As such, a complete closing of the s-wave gap is not necessary (nor observed) because one type of pairing replaces the other. This also agrees with the experimental fact that the apparent critical field for the s-wave phase (referred to as $B_c^{s\text{-wave}} \approx 0.25\text{T}$ in Fig. 3) is larger than B^* . The superconductor becomes 'fully p-wave' at fields well above B^* , in the example of Fig. 3a at $B \sim 0.4\text{T}$ or $\sim 2B^*$. In the range between B^* and about $2B^*$ the two phases coexist which can also explain that the tunnelling spectra in this field range can be described equally well by both the Maki- and the nodal model.

We are really grateful to the Reviewer for raising this important point and did our best to clarify the wording in the revised manuscript.

6. Even for an s-wave superconducting with the reported thickness and T_c , the value of $B^=0.2\text{T}$ for closing the gap seems quite low. The authors did not discuss the fitted depairing strength according to Maki's theory. Is it compatible with the parameters for PdBi_2 ?*

We are grateful to the Reviewer for raising this interesting point; it was not discussed in the original manuscript and is in fact quite subtle. It would be incorrect to treat the extrapolated value of $B_c^{s\text{-wave}} \approx 0.25\text{T}$ in Fig. 3 as a true critical field for s-wave superconductivity because of the existence of the transition to p-wave pairing. A possible explanation of the observed behavior is that the interaction between the two pairings in in-plane fields suppresses s-wave superconductivity at a much lower field than what would be the case without it (or what is observed in the out-of-plane B, where the material remains superconducting up to $\sim 1\text{T}$, Fig. 2d). We have explained this in the revised manuscript and now refer to B^* as a transition (rather than critical) field.